# Distinct Mechanisms of Human Retinal Endothelial Barrier Modulation In Vitro by Mediators of Diabetes and Uveitis

**DOI:** 10.3390/life12010033

**Published:** 2021-12-27

**Authors:** Madhuri Rudraraju, S. Priya Narayanan, Payaningal R. Somanath

**Affiliations:** 1Clinical and Experimental Therapeutics, College of Pharmacy, University of Georgia, Augusta, GA 30912, USA; mrudraraju@augusta.edu; 2Research Division, Charlie Norwood VA Medical Center, Augusta, GA 30912, USA; 3Vascular Biology Center, Medical College of Georgia, Augusta University, Augusta, GA 30912, USA; 4Vision Discovery Institute, Medical College of Georgia, Augusta University, Augusta, GA 30912, USA

**Keywords:** blood–brain barrier, claudin-5, AGE, TNFα, hyperglycemia, lipopolysaccharide

## Abstract

Ocular diseases such as diabetic retinopathy (DR) and uveitis are associated with injury to the blood–retinal barrier (BRB). Whereas high glucose (HG) and advanced glycation end products (AGE) contribute to DR, bacterial infections causing uveitis are triggered by endotoxins such as lipopolysaccharide (LPS). It is unclear how HG, AGE, and LPS affect human retinal endothelial cell (HREC) junctions. Moreover, tumor necrosis factor-α (TNFα) is elevated in both DR and ocular infections. In the current study, we determined the direct effects of HG, AGE, TNFα, and LPS on the expression and intracellular distribution of claudin-5, VE-cadherin, and β-catenin in HRECs and how these mediators affect Akt and P38 MAP kinase that have been implicated in ocular pathologies. In our results, whereas HG, AGE, and TNFα activated both Akt and P38 MAPK, LPS treatment suppressed Akt but increased P38 MAPK phosphorylation. Furthermore, while treatment with AGE and HG increased cell-junction protein expression in HRECs, LPS elicited a paradoxical effect. By contrast, when HG treatment increased HREC-barrier resistance, AGE and LPS stimulation compromised it, and TNFα had no effect. Together, our results demonstrated the differential effects of the mediators of diabetes and infection on HREC-barrier modulation leading to BRB injury.

## 1. Introduction

According to the National Eye Institute, ~14.6 million Americans are expected to have diabetic retinopathy (DR) by 2050, the major cause of vision impairment in the US [1]. A major event in the course of DR is the breakdown of the inner blood–retinal barrier (BRB) composed of retinal endothelial cells (RECs), pericytes, glia, and the outer BRB composed of choriocapillaris and retinal pigment epithelial cells [2,3,4,5]. Retinal ECs and epithelial cells are sealed by the adherens junctions (AJs) and tight junctions (TJs) to form a highly selective barrier for gases, fluids, proteins, lipids, ions, salts, and other nutrients, and inflammatory cells to pass through it by controlled junctional complex alterations and turnover of proteins such as cadherins, claudins, and zona occludens-1 and 2 (ZO-1/2) [6,7]. The characteristics of a particular cell–cell junction differ with the type of cells involved and the type of blood–tissue barrier [8,9,10]. The different repertoire of cell-junction proteins in various vascular beds explains their physiological or pathological standing [5,11,12]. The molecular composition of BRB and their modulation in vision disorders such as DR, retinopathy of prematurity, and uveitis are only starting to emerge.

DR is a neurovascular disease associated with REC dysfunction [13]. Chronic plasma high glucose (HG) contributes to microvascular damage via reduced perfusion and ischemia through several hypoxia-driven factors [14]. The differences in the expression profiles of proangiogenic factors, cytokines, and TJ proteins seem to be dictated by the origin of ECs relating to their vascular bed [15,16]. How the human RECs (HRECs) respond to hyperglycemia and tumor necrosis factor-α (TNFα) are different from bovine retinal ECs, and ECs from other vascular beds such as the human brain and umbilical vein (HUVECs) [17,18] have been reported. Whereas HG suppresses proliferation of HUVECs [19], human pulmonary artery ECs [20], human dermal microvascular ECs [21], porcine aortic ECs [22], and bovine retinal ECs [23], it induces proliferation of HRECs [24,25,26], suggesting that HREC-junctions may have a distinct molecular architecture compared to other vascular beds, which may have implications in various human retinal disorders.

Retinal vascular hyperpermeability occurs through various stimuli, leading to edema and breakdown of BRB [27]. Although high glucose (HG), advanced glycation end products (AGE), TNFα, and lipopolysaccharide (LPS) are known to be modulating the BRB, whether all these directly modulate HREC-barrier is unclear. Since the vascular endothelial growth factor (VEGF) is not the only growth factor that is dysregulated in DR and uveitis, we investigated the modulations in HREC-barrier in vitro and the underlying molecular mechanisms in response to HG, TNFα, bovine serum albumin (BSA)-bound AGE, and LPS to mimic diabetes- and infection-induced inflammation in the human retina. Specifically, we determined the effect of these stimuli on HREC barrier integrity, AJ and TJ protein expression, their intracellular localization, and changes in the phosphorylation of Akt and P38 MAP kinase. Our results reveal distinct mechanisms of HREC-barrier regulation by various agents that are upregulated in diabetes and infection. Treatment with Akt inhibitor triciribine (TCBN) significantly reversed the adverse effects of HG and AGE on the HREC barrier, thus suggesting the potential benefits of TCBN to treat DR-associated BRB injury.

## 2. Materials and Methods

### 2.1. Cell Culture and Reagents

HRECs were maintained in Endothelial Cell Basal Medium fortified with EC growth supplements, antibiotics, and fetal bovine serum (Cell biologics, Chicago, IL, USA). Cells were grown on flasks coated with gelatin (0.2%). Cells from passages 6–10 were used for the experiment. All other plastic culture wares, reagents, and chemicals were purchased from Fisher Scientific, Hampton, NH, USA. Considering that the culture medium already contains 5.5 mM glucose, additional D-glucose was added to prepare a total of 30 mM glucose-containing medium for the HG treatment [28]. The normal medium was served as a control. Primary HRECs were exposed to glucose in two different conditions: (1) by replacing 50 % (50–50) fresh medium each day for 5 d in both the control and treatment groups, and (2) by treating cells once with HG medium with no change for 5 d. BSA-AGE (Cat No. 121800) was purchased from Millipore (Boston, MA, USA). Human TNFα (Cat No. A42550) was purchased from Thermo Scientific (Frederick, MD, USA). LPS lyophilized powder (Cat No. L2018) was obtained from Millipore Sigma (Boston, MA, USA). Triciribine (TCBN) was obtained from Selleckchem (Cat No. S1117), Houston, TX, USA.

### 2.2. Western Blot Analysis

The cell lysates were prepared using immunoprecipitation assay lysis buffer (Millipore, Burlington, MA, USA) supplemented with protease and phosphatase inhibitors (Roche, Basel, Switzerland). Protein concentration was measured by the DC protein assay (Bio-Rad, Hercules, CA, USA), and approximately 30–40 μg of proteins in Laemmli buffer were used. Western blotting was performed as described previously [29]. Densitometry was performed using NIH ImageJ software. Antibodies used include pSer473-Akt (Cat No. 9271), pThr308-Akt (Cat No. 9275), Pan Akt (Cat No. 4685), pP38 MAP Kinase (Cat No. 9215), total P38 MAP Kinase (Cat No. 9212), VE-cadherin (Cat No. 2500), and GAPDH (Cat No. 5174), were purchased from Cell Signaling, Danvers, MA, USA. Anti-Claudin-5 (CLDN5) (Cat No. 352500) antibody was purchased from Thermo Scientific, Waltham, MA, USA. Anti-mouse (Cat No. 170-6516) and anti-rabbit (Cat No. 170-6515) HRP-conjugated secondary antibodies were obtained from Bio-Rad (Hercules, CA, USA). Alexa 485-conjugated secondary antibodies were purchased from Thermo Scientific (Frederick, MD, USA). Antibody dilutions are provided in Appendix A.

### 2.3. Measurement of Endothelial-Barrier Resistance

Endothelial-barrier integrity was measured as the electrical resistance of the endothelial monolayer using electric cell-substrate impedance sensing (ECIS) equipment (Applied Biophysics, Troy, NY, USA) as described previously [30]. To synchronize the HREC monolayer before the treatments, we allowed 24 h of stabilization time to achieve a stable resistance. Once stable resistance was reached, cells were subjected to treatment with insults (30 mM HG [28], 50 µg/mL AGE [31], 10 ng/mL TNFα [32], and 0.1 µg/mL LPS [33,34]) for 24 h or as specified, and the endothelial-barrier resistance was measured in real-time at multiple frequency modes.

### 2.4. Immunofluorescence Staining and Confocal Imaging

Immunofluorescent staining of HREC monolayers was performed in eight-well chamber slides (Fisher Scientific, Hampton, NH, USA). Confluent monolayers were treated for 24 h and washed three times with PBS. The cells were fixed by incubating in ice-cold 4% paraformaldehyde (Cat N. AAJ19943K2) for 20 min followed by PBS wash. The cells were permeabilized with 0.2% Triton X-100 for 15 min and washed with PBS. Cells were incubated in a blocking solution (10% normal donkey serum +0.5% Triton-X in PBS) for one hour. The cell monolayers were then thoroughly washed with PBS before incubation with primary antibodies against CLDN5 (1:200, mouse) and β-catenin (1:100, rabbit) in blocking solution at 4 °C overnight. Immunofluorescence was revealed by incubating in Alexa-Flour secondary antibodies the next day (1:500 dilution of goat anti-rabbit 488 and goat anti-mouse 488) obtained from Thermo Scientific (Waltham, MA, USA). Cells were mounted onto a glass slide using DAPI containing mounting medium (Vector Laboratories, Burlingame, CA, USA). Samples were observed under a confocal microscope equipped with argon and helium/neon lasers (Zeiss, Oberkochen, Germany). Negative controls had just the secondary antibodieswith primary antibodies omitted. All negative controls had no detectable non-specific labeling.

### 2.5. Statistical Analysis

All of the data are presented as mean ± SEM. The “*n*” value for each figure implies the number of samples in each group. All band densitometry analyses are presented as fold changes compared to respective control groups. All of the data were analyzed by parametric testing using Student’s unpaired *t*-test or one-way analysis of variance, followed by the post hoc test (Dunnett’s method) using the GraphPad Prism 6.01 software. Data with *p* < 0.05 were considered significant.

## 3. Results

### 3.1. Treatment with HG Increases CLDN5 Expression

Treatment with HG (50–50) and HG (no media change) for 5 d resulted in the differential regulation of Akt and P38 MAPK activities (Figure 1). Whereas HG (50% media replacement every 24 h) treatment resulted in the increased phosphorylation (both at Ser473 and Thr308) of Akt (Figure 1B–E) and decreased phosphorylation of P38 MPAK (Threonine-180/Tyrosine-182) levels (Figure 1F,G), HG (no change) treatment effects of HRECs were paradoxical. Interestingly, both the HG conditions significantly increased CLDN5 expression in HRECs (Figure 2A–C). Although there was a trend in the upregulated expression of VE-cadherin with either of the HG treatments in HRECs, the data were not significant (Figure 2D,E).

### 3.2. TNFα Induces Modest Changes in Cell-Junction Protein Expression in HRECs

Upon treatment of primary HRECs with TNFα at 10 ng/mL for 24 h, there was a significant increase in the phosphorylation of Akt (Figure 3B,C) and P38 MAPK (Figure 3D). Interestingly, CLDN5 expression was modestly but significantly reduced by TNFα treatment in HRECs (Figure 3E) but the expression of VE-cadherin was elevated considerably (Figure 3F).

### 3.3. Treatment with BSA-Bound AGE Modulates CLDN5 Expression in HRECs

Primary HRECs upon treatment with BSA-bound AGE showed a dose-dependent increase in the phosphorylation of Akt (Figure 4A–C) and P38 MAPK (Figure 4D). Interestingly, although the CLDN5 expression appeared to be increased with 25 and 50 µg/mL AGE, we observed a sharp decrease in CLDN5 expression with 100 µg/mL treatment (Figure 4E). Intriguingly, the expression of VE-cadherin was increased with 100 µg/mL of AGE treatment (Figure 4F) with no significant effect on other doses.

### 3.4. LPS Treatment Suppresses CLDN5 Expression with No Changes in VE-Cadherin

Treatment of HRECs with LPS resulted in significantly reduced Akt phosphorylation at both Serine-473 and Threonine-308 residues (Figure 5A–C) associated with increased P38 MAPK (Threonine-180/Tyrosine-182) phosphorylation (Figure 5A,D). LPS treatment for 24 h robustly decreased CLDN5 expression (Figure 5A,E) with a modest increase in VE-cadherin expression (Figure 5A–F) specifically when treated with 0.1 µg/mL dose for 24 h.

### 3.5. Modulators of Diabetes and Infection Associated Inflammation Have Distinct Effects and Mechanisms to Modulate the HREC Barrier

Measurement of HREC barrier resistance by ECIS assay revealed an increase in monolayer barrier strengthening with HG treatment for 24 h (Figure 6A). HREC monolayers treated with TNFα for 24 h displayed no significant changes in barrier resistance in an ECIS assay (Figure 6A). In comparison to TNFα, treatment with LPS showed a substantial reduction in HREC-barrier resistance compared to the control (Figure 6A). Like HG, treatment of HREC monolayers with 50 µg/mL dose of AGE exhibited a significant reduction in barrier resistance compared to control (Figure 6B).

Since we specifically saw increased activation of Akt in HRECs treated with HG and AGE, we determined the effect of TCBN, an Akt inhibitor, on reversing the impact of HG and AGE on HREC monolayers and cell-barrier normalization in a 24 h ECIS assay. Co-treatment of HRECs with TCBN prevented the loss of AGE (50 μg/mL)-induced barrier resistance (Figure 6B). Interestingly, treatment of HREC monolayers with TCBN alone had a significant effect in increasing barrier resistance. Western blot analysis of AGE-Treatment of HRECs showed a consistent increase in Akt phosphorylation at Threonine-308 residues (Figure 6C,D), which was reversed upon co-treatment with 10 μM TCBN [29,30,35,36].

Next, we treated HREC monolayers with HG, AGE, TNFα, TCBN, and LPS for 24 h, and performed immunostaining analysis to study the intracellular distribution of CLDN5 (TJ) and β-catenin (AJ). In our analysis, treatment with TNFα for 24 h resulted in an abnormal pattern of CLDN5 distribution in HRECs. The loss of CLDN5 in HREC-junctions was evident with LPS treatment (Figure 7). Interestingly, although TCBN treatment indicated reduced CLDN5 distribution in HREC junctions, there were no developed gaps visible in the HREC junctions, indicating that, despite reduced CLDN5 in cell junctions, TCBN treatment increased HREC-barrier resistance as determined by the ECIS analysis. In contrast to LPS but in agreement with the Western blot results, treatment with AGE and HG resulted in increased CLDN5 expression but its distribution altered, both of which were reversed and normalized by co-treatment with TCBN (Figure 7).

Apart from the TJ proteins, we determined the changes in β-catenin localization in HREC monolayers post-treatment with HG, AGE, TNFα, TCBN, and LPS 24 h. Treatment with TNFα showed an intact AJ with normal β-catenin localization in the cell–cell junctions (Figure 8). By contrast, treatment with LPS exhibited reduced β-catenin localization and increased gap formation in the HREC-barrier junctions, but no changes were observed in β-catenin localization with TCBN treatment. Interestingly, treatments with AGE and HG resulted in altered localization of β-catenin in HREC-barrier junctions, which was reversed and normalized by co-treatment with TCBN (Figure 8). Together, these results demonstrate distinct effects of HG, AGE, TNFα, and LPS and Akt inhibitor TCBN on HREC-barrier junctions.

## 4. Discussion

Regardless of the etiology and molecular mechanisms, vision disorders are associated with injury to the BRB, recurrent episodes of inflammation, vascular permeability, and edema [3,37]. Whereas BRB damage in DR is driven by hyperglycemia, AGE, and VEGF [27], bacterial infections injure the BRB through the endotoxin-mediated EC and epithelial cell damage [33,38].

In both conditions, TNFα plays a significant role in inducing inflammation, both pre- and post-tissue injury [39]. Unlike the endotoxin-induced BRB injury and vascular permeability in pathologies such as uveitis, BRB damage in DR is biphasic, characterized by an early vascular loss in the retina followed by hypoxia-induced abnormal neovascularization [40], causing vascular hyperpermeability and tuft formation in the proliferative stages of DR [40]. The underlying molecular mechanisms of various vision disorders, specifically on the direct modulators in HREC-barrier protein expression, remain largely unclear. There are no existing reports on how LPS affects the Akt pathway and barrier junction protein expression in HRECs. In the current study, we demonstrated that mediators of diabetes such as HG and AGE modulate HREC-barrier function, cell-junction protein turnover, their intracellular distribution, and phosphorylation of Akt and P38 MAPK differently compared to treatment with bacterial LPS and TNFα.

Despite the rich literature on the effect of HG on ECs in vitro, the outcome varies based on the context, species, vascular bed, micro- vs. macrovascular, and the experimental conditions such as dose, duration, and frequency of treatments, and media change, etc. HG inhibits migration [41,42], and induces apoptosis [43,44,45] and permeability [46] in various cell types. In ECs, HG suppresses proliferation in HUVEC [19], human pulmonary artery ECs [20], human dermal microvascular ECs [21], porcine aortic ECs [22], and bovine RECs [23]. Conversely, there have been conflicting reports on REC response to HG. Several studies have reported enhanced mouse REC-migration when exposed to HG with no effects on apoptosis and capillary morphogenesis [47]. Intriguingly, studies have also reported HG-induced REC proliferation [24,25,26] as a mechanism driving the vascular proliferative stage of DR [48]. While HG treatment of HUVEC resulted in increased P38 MAPK activity and suppressed proliferation [49], P38 MAPK activity was not affected, and VEGF expression was unaltered when RECs were cultured in HG medium [47]. In our analysis, HG treatments elicited distinct responses in two different conditions. Whereas five days of HREC treatment with HG with no media change suppressed Akt and activated P38 MAPK, replenishing 50% of the HG-containing media every 24 h activated Akt and suppressed P38 MAPK. Surprisingly, in both conditions, HG treatment increased CLDN5 expression. These variations and unaltered HREC-barrier resistance by HG in our ECIS analysis suggest that further changes in experimental conditions may be required in modeling HG as a suitable in vitro model of DR.

Although alterations in several pathways coincide with the physiological and pathological cell-barrier modulations, protein turnover in AJs and TJs is primarily regulated by the Src and Akt pathways, respectively, with some mutual cross-talk [5,12]. Activation of P38 MAPK that drives the cell stress and inflammation [50] has also been reported to modulate AJs via the GSK3-β-catenin pathway [51,52]. Src-mediated VE-cadherin internalization and AJ breakdown promote short-term vascular permeability by VEGF [29]. Interestingly, long-term activation of Akt by VEGF is essential to restore the barrier function, which is achieved through AJ stabilization by inhibiting the GSK3-β-catenin pathway and TJ stabilization by transcriptional upregulation of CLDNs [29,30]. Sustained Src or P38 MAPK activation and chronic inhibition of Akt lead to EC dysfunction [53]. Although it is known that Akt inhibition will result in the loss of barrier integrity [36], overexpression of Akt (myrAkt) has also proven to cause retinal vascular malformations [54]. Activation of the PI3K/Akt pathway activity has been indicated to promote HG-induced ECM secretion and cellular hypertrophy [55,56]. The enhanced vascular tuft formation in DR could be due to the hyperactivation of Akt and overexpression of TJ proteins coinciding with abnormal neovascularization, together causing vascular malformations. Since consistent Akt activation with HG and AGE in diabetes has been demonstrated [57,58], targeting Akt was suggested as a therapeutic strategy to treat DR [59]. This suggests that a fine-tuning of Akt, as well as AJ and TJ protein expression, may be crucial in retinal vasculature physiology. Alluding to this, we observed hyperactivation of Akt increased expression and junctional accumulation of CLDN5 with AGE and HG treatment in HRECs, and impaired HREC barrier-resistance in vitro by AGE. The effect of the Akt inhibitor, TCBN, to reverse the adverse effects of HG and AGE on HRECs further supports this view.

The next obvious question would be an alternative of a suitable in vitro model to study diabetic complications in RECs. Although the HG is a preferred model, like any other experimental model, HG also comes with some limitations such as different conditions used by various laboratories, and discrepancies in the endpoints associated with it. In the retina, AGE induces oxidative stress and inflammation to promote vascular dysfunction [60]. Intravenous administration of AGEs in non-diabetic rats led to their accumulation in the retinal blood vessels and induced pathophysiological hallmarks of DR such as mitochondrial swelling, thickening of the retinal basement membrane, and pericyte loss, etc. [61]. Deposition of AGE adducts in mice induced REC-barrier disruption and retinal capillary hyperpermeability [31,62] through VEGF production [63], suggesting AGE might be a better candidate to mimic DR effects in ECs in culture. Agreeing with this, we observed impaired barrier resistance associated with the activation of Akt and P38 MAPK and increased CLDN5 expression in HRECs with AGE treatment.

Although inflammation and vascular permeability occur in DR and uveitis, the retinal vascular pathology and the involvement of EC-junctions in each condition appear to be quite different. Furthermore, proinflammatory cytokines such as TNFα are elevated in both conditions [64,65]. TNFα has been shown to induce REC permeability by reducing the expression of TJ proteins through activation of PKCζ and NF-κB [32]. Where continuous TNFα is antiangiogenic, a pulse of TNFα has been shown to prime ECs and is proangiogenic [66]. By contrast, another study reported that a TNFα increase did not contribute to BRB breakdown in early DR but led to BRB breakdown at later time points, suggesting that the BRB loss in the advanced DR could be a result of other factors, such as advanced lipoxidation end products (ALE) [67], AGE and VEGF [68]. In our analysis, we did not see any significant effect of TNFα on HREC-barrier protein modulation or monolayer resistance. Interestingly, we observed abnormal distribution of CLDN5 but not β-catenin with TNFα treatment on HREC monolayers, however, with an intact cell-barrier. Together, our findings suggest that the role of TNFα is likely to prepare the HREC monolayer for inflammation rather than directly modulating the cell junctions.

In contrast to HG and AGE, treatment of RECs with LPS resulted in reduced expression of VE-cadherin and CLDN5, increased gap formation in the monolayers, and reduced monolayer barrier resistance. The literature indicates that knocking down TLR4, a receptor for LPS [69], results in increased ZO-1 and occludin levels in RECs subjected to HG [70], implying that TLR4 may be directly involved in REC-barrier regulation [34,71]. The effect of LPS on Akt suppression and CLDN5 downregulation has also been demonstrated in human lung EC cells [35]. In our analysis, we also observed a decrease in Akt phosphorylation in HRECs with LPS treatment, the possible reason for decreased CLDN5 expression.

In summary, the current study offers novel insights into the different molecular mechanisms by which HG, AGE, TNFα, and LPS modulate primary HREC-junction protein turnover, their distribution within the cells, and the monolayer barrier resistance in vivo. Since most studies until today have been focused on the LPS-induced inflammation in the retina, the current study is a first report on how LPS modulates claudin-5, VE-cadherin, and β-catenin in HRECs. Furthermore, our study indicates the need for a fine tuning in the Akt activity and HREC-junction protein expression to maintain barrier integrity. However, the study comes with a few limitations, the major one being that this is an in vitro study, and hence does not simulate perfect conditions of the disease in vivo. It is likely possible that each of the stimuli used in our study may elicit a different response in vivo, where they mediate their effects in conjunction with other molecules and conditions. Nevertheless, the study has laid a foundation to further explore the mechanisms leading to retinal diseases such as DR and uveitis, a reliable model in AGE to mimic diabetes effects on HRECs, and a potential candidate in TCBN to treat proliferative DR. Single-cell sequencing [72,73] of HRECs treated with various growth factors and cytokines in the future will yield more reliable information to help with therapeutic development for vision disorders.

## Figures and Tables

**Figure 1 life-12-00033-f001:**
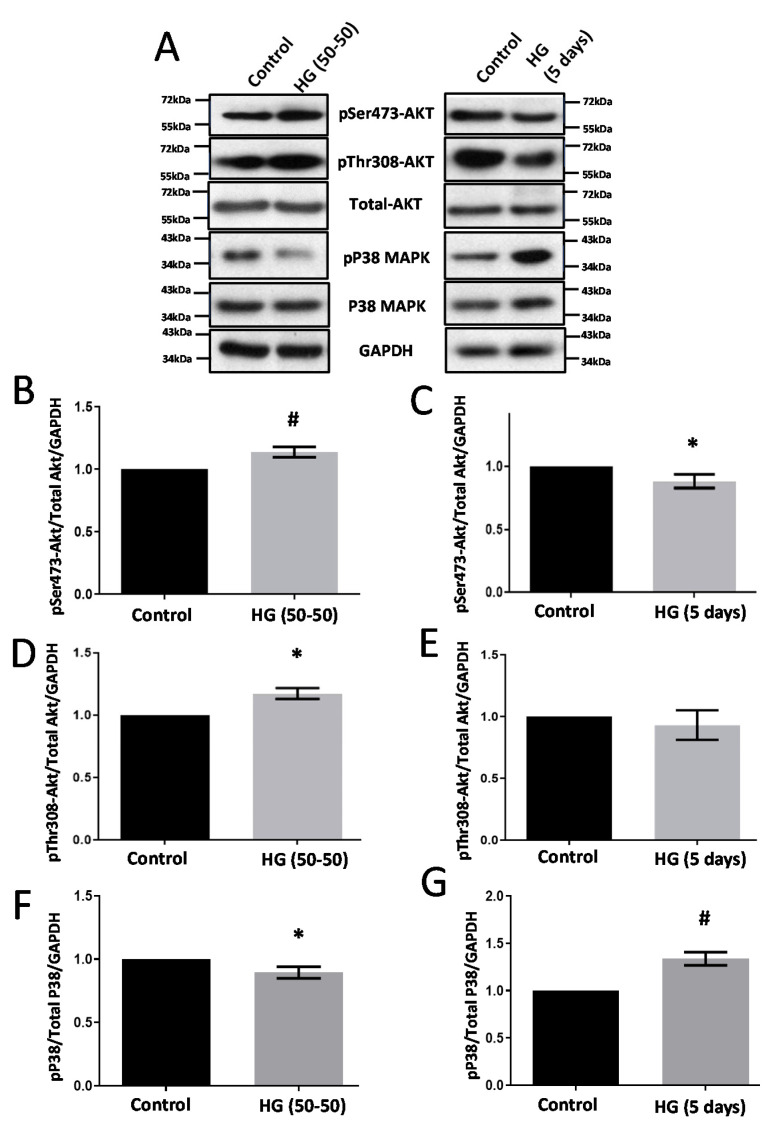
HG treatments in two different conditions differentially modulate Akt and P38 MAPK phosphorylation in HRECs. Two conditions of HG treatments (50–50 change every 24 h versus no change in media for (5 d) were performed in HREC monolayers. (**A**) Representative Western blot images of HREC lysates treated with two different conditions of HG (30 mM) showing changes in the expression of phosphorylated Akt and P38 MAP kinase compared to their respective controls. (**B**,**C**) Bar graph showing band densitometry analysis of pSer473-Akt expression in HRECs treated with HG (50–50 (*n* = 6) versus no change (*n* = 14), (5 d) compared to respective untreated controls. (**D**,**E**) Bar graph showing band densitometry analysis of pThr308-Akt expression in HRECs treated with HG (50–50 versus (*n* = 3) no change (*n* = 3), (5 d) compared to respective untreated controls. (**F**,**G**) Bar graph showing band densitometry analysis of pP38 MAPK expression in HRECs treated with HG (50–50 (*n* = 6) versus no change (*n* = 16), (5 d) compared to respective untreated controls. Data are presented as Mean + SEM. * *p* < 0.05; # *p* < 0.001.

**Figure 2 life-12-00033-f002:**
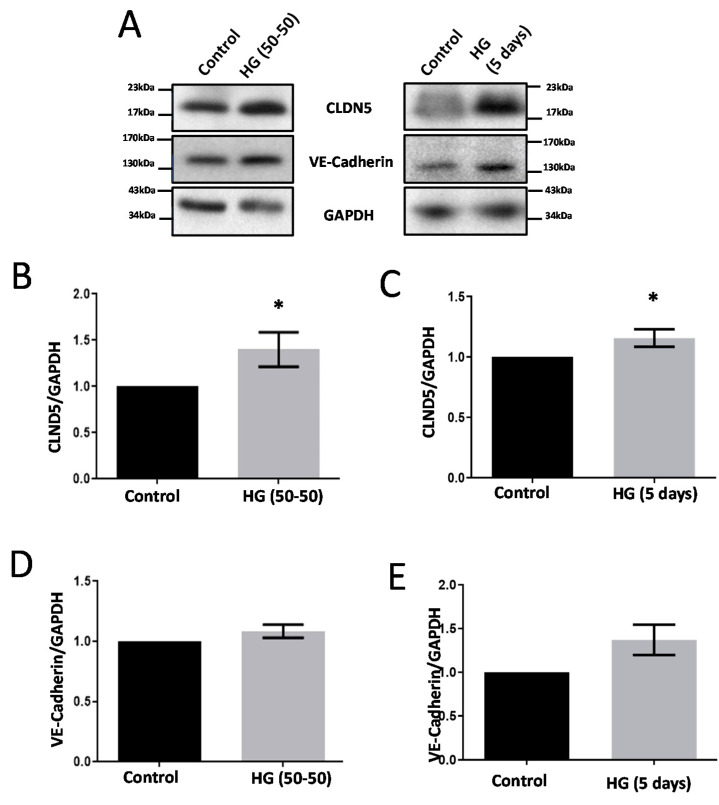
HG treatments in two different conditions differentially modulate HREC-barrier protein expression in vitro. (**A**) Representative Western blot images of HREC lysates treated with two different conditions of HG (30 mM) for 5 d showing changes in the expression of AJ protein VE-cadherin and TJ protein CLDN5. (**B**,**C**) Bar graph showing band densitometry analysis of CLDN5 expression in HRECs treated with HG (50–50 (*n* = 4) versus no change (*n* = 15), (5 d) compared to respective untreated controls. (**D**,**E**) Bar graph showing band densitometry analysis of VE-cadherin expression in HRECs treated with HG (50–50 (*n* = 6) versus no change (*n* = 3), (5 d) compared to respective untreated controls. Data is presented as Mean + SEM. * *p* < 0.05.

**Figure 3 life-12-00033-f003:**
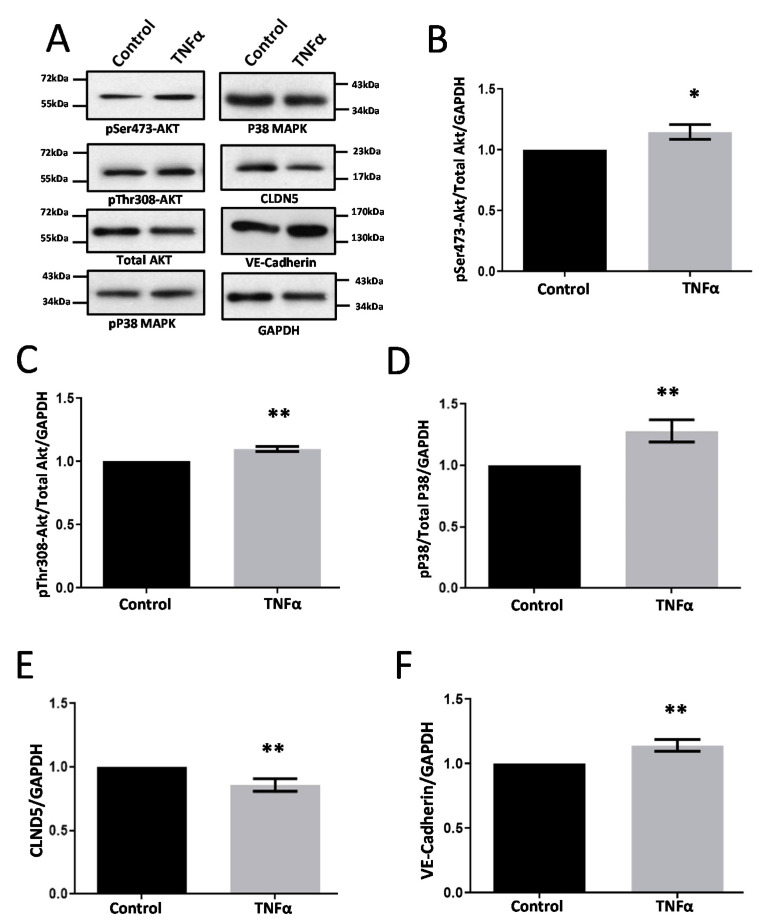
TNFα treatment of HRECs reduces CLDN5 and increases VE-cadherin expression. (**A**) Representative Western blot images of HREC lysates treated with TNFα (10 ng/mL) for 24 h showing changes in the expression of phosphorylated Akt and P38 MAP kinase, VE-cadherin, and CLDN5. (**B**,**C**) Bar graph showing band densitometry analysis of pSer473-Akt (*n* = 5) and pThr308-Akt (*n* = 3) expression in HRECs treated with TNFα compared to untreated controls. (**D**) Bar graph showing band densitometry analysis of pP38 MAPK expression in HRECs treated with TNFα compared to untreated controls (*n* = 14). (**E**,**F**) Bar graph showing band densitometry analysis of CLDN5 (*n* = 12) and VE-cadherin (*n* = 8) expression in HRECs treated with TNFα compared to untreated controls. Data are presented as Mean + SEM. * *p* < 0.05; ** *p* < 0.01.

**Figure 4 life-12-00033-f004:**
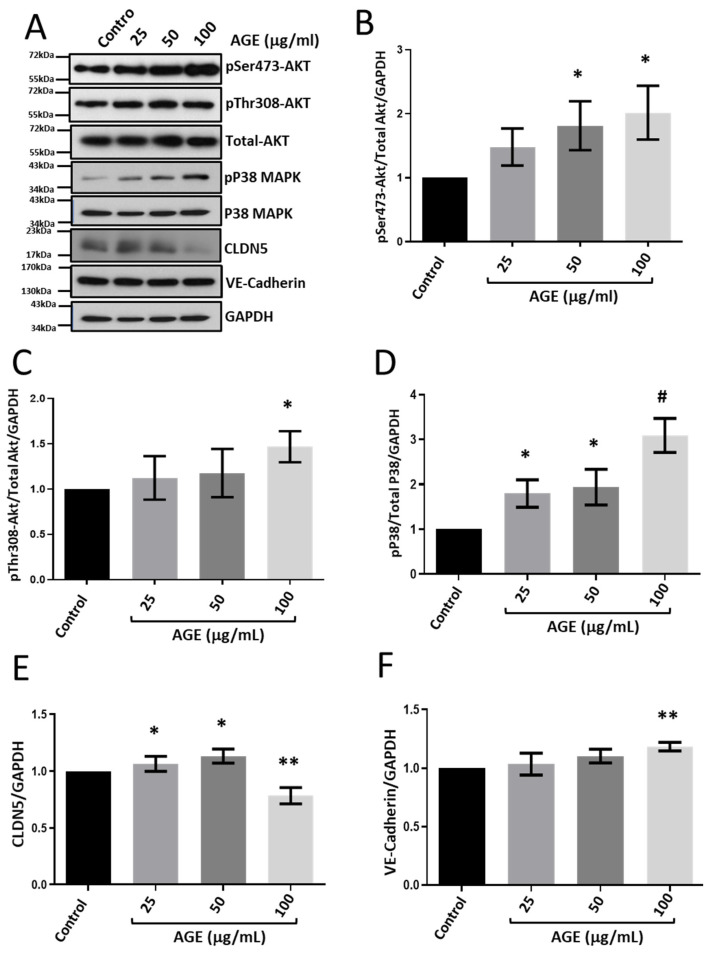
Treatment with AGE activated Akt and P38 MAPK and increased VE-cadherin expression in HRECs. (**A**) Representative Western blot images of HREC lysates treated with AGE (25, 50, and 100 μg/mL) for 24 h show changes in phosphorylated Akt and P38 MAP kinase expression VE-cadherin, and CLDN5. (**B**,**C**) Bar graph showing band densitometry analysis of pSer473-Akt (*n* = 6) and pThr308-Akt (*n* = 4) expression in HRECs treated with AGE compared to untreated controls. (**D**) Bar graph showing band densitometry analysis of pP38 MAPK expression in HRECs treated with AGE compared to untreated controls (*n* = 6). (**E**,**F**) Bar graph showing band densitometry analysis of CLDN5 (*n* = 5) and VE-cadherin (*n* = 4) expression in HRECs treated with AGE compared to untreated controls. Data are presented as Mean + SEM. * *p* < 0.05; ** *p* < 0.01; # *p* < 0.001.

**Figure 5 life-12-00033-f005:**
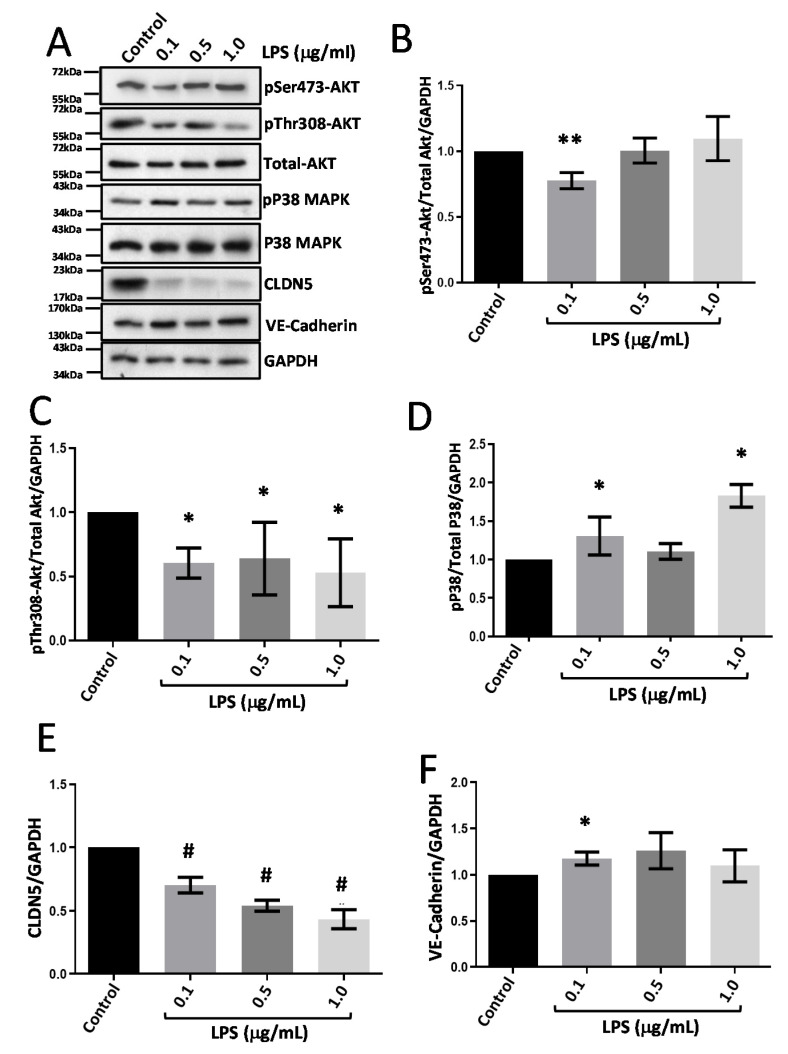
Treatment of HRECs with LPS reduced expression of CLDN5 but not VE-Cadherin. (**A**) Representative Western blot images of HREC lysates treated with LPS (0.1, 0.5, and 1 μg/mL) for 24 h showed changes in phosphorylated Akt and P38 MAP kinase expression VE-cadherin, and CLDN5. (**B**,**C**) Bar graph showing band densitometry analysis of pSer473-Akt (*n* = 6) and pThr308-Akt (*n* = 3) expression in HRECs treated with LPS compared to untreated controls. (**D**) Bar graph showing band densitometry analysis of pP38 MAPK expression in HRECs treated with LPS compared to untreated controls (*n* = 5). (**E**,**F**) Bar graph showing band densitometry analysis of CLDN5 (*n* = 10) and VE-cadherin (*n* = 6) expression in HRECs treated with LPS compared to untreated controls. Data are presented as Mean + SEM. * *p* < 0.05; ** *p* < 0.01; # *p* < 0.001.

**Figure 6 life-12-00033-f006:**
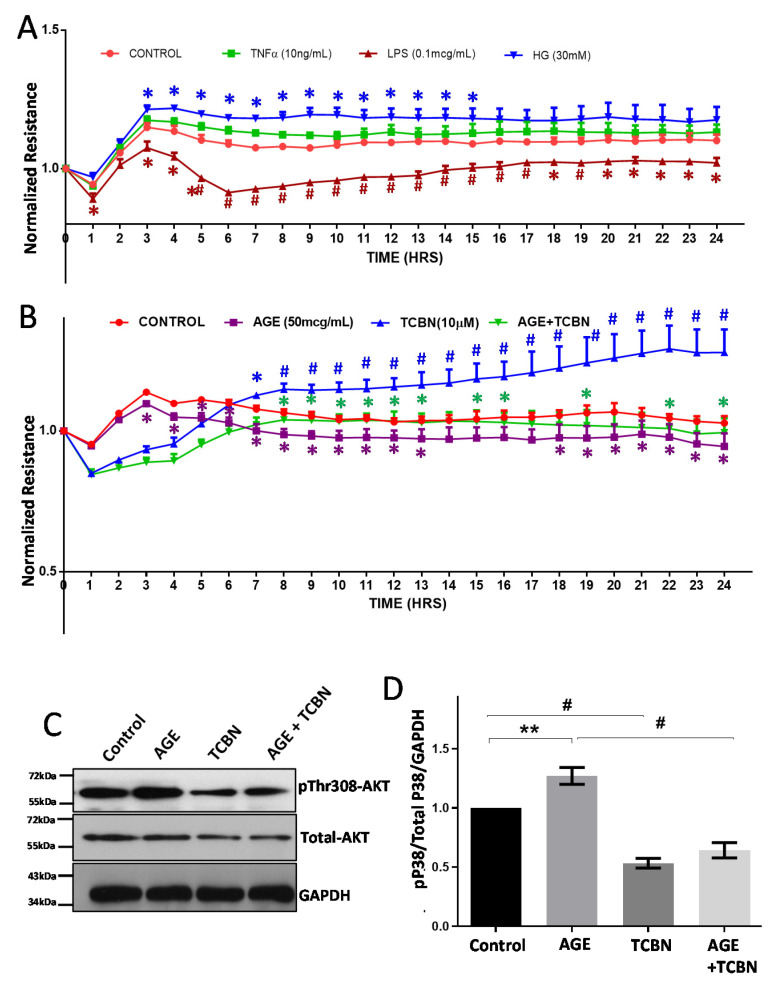
HG, AGE, LPS, and TNFα differentially modulated HREC-barrier resistance. (**A**) Graph showing real-time changes in HREC-barrier resistance after treatments with HG (30 mM), TNFα (10 ng/mL), and LPS (0.1 μg/mL) for 24 h compared to saline-treated control as measured using ECIS equipment (*n* = 4). (**B**) Graph showing real-time changes in HREC-barrier resistance after treatments with 50 μg/mL AGE and/or 10 μM TCBN for 24 h compared to saline-treated control as measured using ECIS equipment (*n* = 4). (**C**,**D**) Representative Western blot images and the densitometry analysis of HRECs treated with 50 μg/mL AGE and/or 10 μM TCBN (Akt inhibitor) for 24 h showing changes in pSer308Akt phosphorylation normalized to total Akt and GAPDH (*n* = 4). Data are presented as Mean + SEM. * *p* < 0.05; ** *p* < 0.01; # *p* < 0.001.

**Figure 7 life-12-00033-f007:**
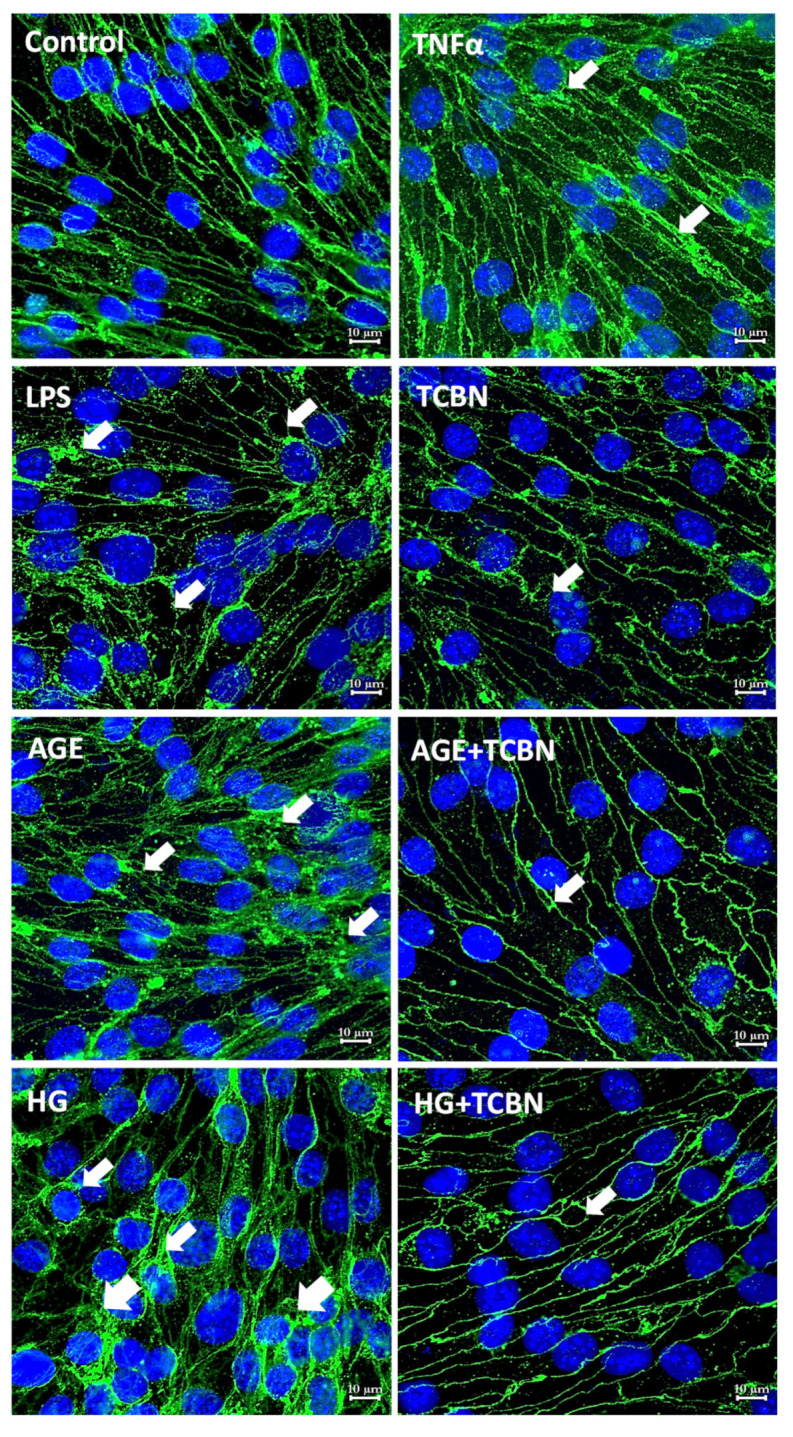
HG, AGE, LPS, and TNFα treatment resulted in different intracellular distribution of CLDN5 in HRECs. Representative confocal images of HREC monolayers subjected to 24 h treatment with TNFα (10 ng/mL), LPS (0.1 μg/mL), AGE (50 μg/mL), HG (30 mM), TCBN (10 μM), HG + TCBN, and AGE + TCBN showing changes in the intracellular distribution of CLDN5 compared to PBS control. Images were taken using an LSM confocal microscope at 63× magnification (Scale bars: 10 μM).

**Figure 8 life-12-00033-f008:**
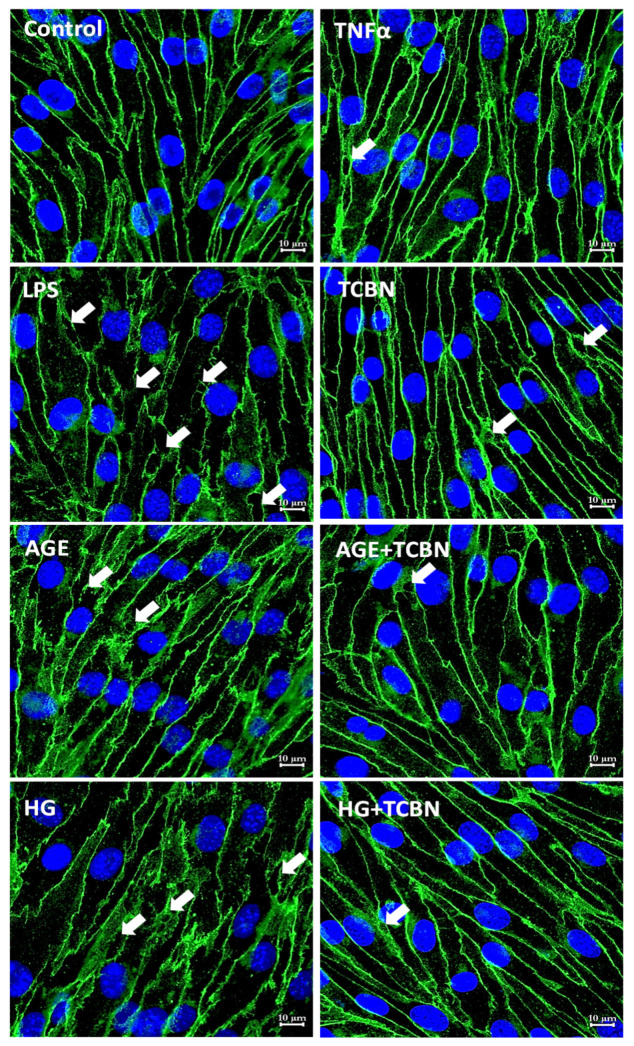
HG, AGE, LPS, and TNFα treatment resulted in different intracellular distribution of β-catenin in HREC AJs. Representative confocal images of HREC monolayers subjected to 24 h treatment with TNFα (10 ng/mL), LPS (0.1 μg/mL), AGE (50 μg/mL), HG (30 mM), TCBN (10 μM), HG + TCBN, and AGE + TCBN showing changes in the intracellular distribution of β-catenin compared to PBS control. Images were taken using an LSM confocal microscope at 63× magnification (Scale bars: 10 μM).

## Data Availability

All the data are included in the manuscript.

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
