# Peer review of "Distinct Mechanisms of Human Retinal Endothelial Barrier Modulation In Vitro by Mediators of Diabetes and Uveitis"

_life, 2021, doi:10.3390/life12010033_

Round 1

Reviewer 1 Report

This manuscript describes in vitro data about how high glucose, advanced glycation end products, TNFα and LPS affect Akt and p38 MAPK signaling, expression and subcellular localization of Claudin 5 and VE-Cadherin, and barrier resistance of human retinal endothelial cells. The data presented here is clinically relevant and the manuscript is well written. The Authors also provide mechanistic data to explain the described observations. However, a few points should be clarified before the final publication of the manuscript:

Specific suggestions:

  1. It should be included in the title that the observed phenomenon was found in vitro.
  2. Working dilutions of the primary antibodies should be included in 2.2.
  3. In Figures 1A, 2A, 3A, 4A, 5A and 6B, molecular weight markers should be provided.
  4. The order of subfigures in Figure 6 does not follow the flow of the manuscript text.
  5. In Figures 7 and 8, arrows should point to the major characteristic changes described in the manuscript text.
  6. All abbreviations should be given when first mentioned in the text.
  7. There are misspellings in Figures 3A and 4E.

Author Response

We thank the reviewer for outlining the strengths of our study. We have addressed all the concerns as delineated below:

  • The title has been modified to reflect ‘in vitro’ nature of the study.
  • A separate Appendix Table A has been added to the manuscript with the working dilutions of the primary antibodies.
  • Molecular weight markers have been provided to all the Western blot images in the manuscript.
  • Figure 6 panels (and legends) are re-aligned to match the flow of information in the text.
  • All abbreviations have been re-checked and confirmed to have a first expanded in the text.
  • Misspellings in Figures 3A and 4E have been corrected.

Reviewer 2 Report

Diabetic retinopathy (DR) is a neurovascular disease associated with retinal endothelial cells (REC) dysfunction, which is the major component of the blood-retinal barrier (BRB). High glucose (HG) and advanced glycation end products (AGE) cause the permeabilization of BRB followed by the damage of BRB. In this article, Madhuri Rudraraju et. al have elucidated distinct mechanisms of HREC-barrier regulation and permeability by various agents that are upregulated in diabetes and infection. Furthermore, the authors elucidated that treatment with Akt inhibitor triciribine (TCBN) significantly reversed the adverse effects of HG and AGE on the HREC barrier, thus suggesting the potential benefits of TCBN to treat BRB injury associated with DR.

On the other hand, HG and TNFα treatments increased the MAPK phosphorylation a minimal, even if significantly (Figures 1-3). In Figure 4, AGE elicited the activation of ATK and P38 MAPK more than 2-folds, but were AGE concentrations compatible to its levels in diabetic patients? The authors had better discuss whether MAPK phosphorylation stream is the major pathway of the BRB deterioration. Is this mechanism a novel pathway in BRB deterioration? In summary, the authors highlighted the need for a ‘fine tuning’ in the Akt activity and HREC-junction protein expression to maintain barrier integrity. They hypothesized that it is likely possible that each of the stimuli used in their study may elicit a different response in vivo, where they mediate their effects in conjunction with other molecules and conditions. The readers would like to know the detail of the deterioration mechanisms of BRB. The indication by the authors is promising and the therapeutic approach can lead to enable future design and development of novel therapeutic agents to minimize and restore BRB damage, because they demonstrated that treatment with Akt inhibitor triciribine (TCBN) significantly reversed the adverse effects of HG and AGE on the HREC barrier.

Author Response

The authors thank the reviewer for the positive comments on our study. Below you will find our point-by-point response to the critique:

  • The clinical comparison of the dose of AGE for diabetic patients is a great question. Unfortunately, there is no clarity on the clinical levels of AGE in diabetic patients due to high personal variations and different stages of the disease. Hence, we looked at the already standardized range of concentrations for in vitro analysis from the published papers (references are included). Based on the dose-response we observed, which is in agreement with the literature, we picked the optimal dose of 50 micrograms for stimuli.
  • The reviewer is correct that MAPK phosphorylation and activation is a major pathway in BRB deterioration. This is not a novel mechanism as the pathway has been well characterized and established in BRB deterioration. Hence, in the current study, we only used it as a marker to determine the direct effects of different stimuli. We have discussed the P38MAPK role in BRB on pages 11 and 12.

Reviewer 3 Report

In this research article, Rudraraju et.al., investigates different molecular mechanisms by which HG, AGE, TNFα, and LPS modulate primary HREC-junction protein turnover and their distribution within the cells. This article is a valuable addition to the field as it demonstrates the differential effects of the mediators of diabetes and infection on HREC-barrier modulation leading to BRB injury, along with potential candidate in TCBN to treat BRB injury. This article has interesting observations which are beneficial to researchers in the areas of DR, Uveitis, and other retinal associated complications. However, the manuscript is quite confusing in parts, with figures, syntax, and spacing needs to be revised in several sections with a thorough proof check of the manuscript.

Some the other major concerns are discussed below.

Introduction

-Line 33: Add more information on how highly selective barrier is controlled by alterations and protein turnover.

-Line 54: Clarify the full form of VEGF.

Materials and Methods

-Line 72: Add a reference for using HG 30mM. Also add the information about concentration and treatment time of TNFα, AGE and LPS in this section along with appropriate references. No information is mentioned about Akt inhibitor TCBN in this section and these details are necessary.

-Line 86 – 90: Add the concentrations used for the primary antibodies.

-Line 92: Add the catalogue numbers and concentrations used for secondary antibodies as well.

-Line 115: Add the catalogue number as well.

-Line 144: Add more information about which post hoc test was used.

Results

-Graphs in all the results can be improved by moving the n numbers and *,# explanations to the figure captions. Furthermore, add error bars in both directions on bar graphs.

- Add the molecular weight for the proteins in all the western blot images in results section.

-Figure 1 A: Add the bands for GAPDH which were used quantifications.

-Figure 2 A: Add treatment group labels.

-Figure 4 A: Give an explanation for why bands for β-Actin are shown, when quantification is done by GAPDH loading control.

-Figure 7 and 8: Add some arrows to highlight the different intracellular distributions with different treatments.

Discussion

-Line 385: Also discuss how other factors such as growth factors and lipoxidation (ALEs) might also affect the HREC-barrier modulation, with references from recent papers.

-Furthermore, discuss future directions such as single cell sequencing for future investigations and reference recent papers which used this technique in RECs.

Author Response

The authors thank the reviewer for the detailed review and the positive comments on our study. Below you will find our point-by-point response to the critique:

  • We have described how barrier selectivity is controlled by junctional alterations and protein turnover.
  • VEGF has been expanded.
  • We have added a reference for using HG 30mM.
  • The dose and duration of specific treatments have been separately provided under each experimental procedure as some conditions are different. Hence, to avoid redundancy, we did not include this information again in the specified paragraph. We apologize for not including TCBN in methods, which has been included in the revised manuscript. Once again, treatment details and references for TCBN are separately provided in the experimental details.
  • Concentrations used for the primary antibodies have been included in a separate table. Catalog numbers and dilutions of secondary antibodies, and the reagent have been included as well.
  • Post hoc test information has been included in the statistical analysis section.
  • The n numbers and *, # explanations in the graphs have been moved to the figure captions. Error bars on bar graphs have been included in both directions.
  • Molecular weight markers have been provided to all the mentioned Western blot images.
  • Representative GAPDH bands used for quantifications are included in Figure 1 A.
  • Treatment group labels have been included in Figure 2 A.
  • We apologize, the β-Actin image in Figure 4 A was an error on our end. In fact, all the loading controls used in the study were GAPDH and no β-Actin was used. We have corrected this error.
  • We have provided arrows in Figs. 7 and 8 to highlight treatment-specific intracellular distributions.
  • We have included a brief discussion on the effect of lipoxidation (ALEs) in HREC-barrier modulation with the citation of a recent review article.

We have proposed “single-cell sequencing” with citations as a future strategy in the concluding paragraph.

Round 2

Reviewer 3 Report

Rudraraju et.al., have addressed all the concerns raised in the initial article and the efforts are greatly appreciated. The revised research article can deserve publication without further changes.